# Monitoring the Status of Soil-Transmitted Helminthiases in Non-Endemic Implementation Units: A Case Study of Borgu in Northcentral Nigeria

**DOI:** 10.3390/pathogens12030491

**Published:** 2023-03-21

**Authors:** Babatunde Adewale, Hammed Mogaji, Joshua Balogun, Emmanuel Balogun, Francisca Olamiju, De’Broski Herbert

**Affiliations:** 1Public Health and Epidemiology Department, Nigerian Institute of Medical Research, Yaba Lagos 101245, Nigeria; 2Parasitology and Epidemiology Unit, Department of Animal and Environmental Biology, Federal University Oye-Ekiti, Oye-Ekiti 371104, Nigeria; 3Mission to Save the Helpless, Jos 930001, Nigeria; 4Department of Biological Sciences, Federal University Dutse, Dutse 720223, Nigeria; 5Department of Biochemistry, Ahmadu Bello University, Zaria 810107, Nigeria; 6Department of Pathobiology, School of Veterinary Medicine, University of Pennsylvania, Philadelphia, PA 19104, USA

**Keywords:** soil-transmitted helminthiases, elimination, control, mapping, monitoring, Niger, Nigeria

## Abstract

Nigeria remains the most endemic country in sub-Saharan Africa (SSA) for soil-transmitted helminthiases (STH). In line with ongoing monitoring plans, we present findings from a recent analysis of STH epidemiological data in Borgu, one of the non-endemic implementation units for STH in the northcentral region of Nigeria. An overall prevalence of 8.8% was recorded for STH infection, which corresponds to a 51.9% decline from the 18.3% reported in 2013. All the infected participants (36 out of 410) had a low intensity of infection. However, more than two-thirds (69%) of the children do not have access to latrine facilities, and 45% of them walk barefoot. Prevalence was significantly associated with community, age, and parental occupation. About 21–25% reduced odds were reported in some of the study communities, and children whose parents were traders had 20 times lower odds of infection compared to those whose parents were farmers. The ongoing preventive chemotherapy program for lymphatic filariasis in the area could be responsible for the huge reduction in prevalence and intensity estimates for STH. It is therefore important to invest in monitoring transmission dynamics in other non-endemic areas to arrest emerging threats through the provision of complementary interventions including WASH facilities and other health educational tools.

## 1. Introduction

Soil-transmitted helminthiases (STH) is one of the most common groups among the 20 neglected tropical diseases identified by World Health Organization (WHO) [1]. Over a billion people are estimated to be at risk, with the majority of them living in sub-Saharan Africa (SSA) [1,2]. Infections are commonly found in marginalized settings, where poverty prevails and water, sanitation, and hygiene resources are either lacking or insufficient [3,4]. Most human STH infections are from four parasitic nematode species whose development are dependent on the soil environment: *Ascaris lumbricoides* (Linnaeus, 1978), known as roundworm; *Trichuris trichiura* (Linnaeus, 1771), known as whipworm; and *Necator americanus* (Stiles, 1902) and *Ancylostoma duodenale* (Dubini, 1843), both known as hookworms. [3]. People get infected when they come in contact with contaminated soils harboring the infective stages of these parasites, either through the ingestion of infective eggs for roundworms and whipworms or skin penetration by infective larva of hookworms [3]. Poor hygiene practices and increased mobility have contributed to the vulnerability of children below age 15 to these infections with substantial health and socioeconomic consequences such as malnutrition, anemia, impaired cognition, and the loss of school learning hours [4,5]. 

Periodic large-scale administration of albendazole/mebendazole medicines have been adopted as the control strategy for STH following the recommendations from the WHO [6]. This intervention, known as preventive chemotherapy (PC), is targeted at 75% of children who are between the ages 5 and 14 years [6,7] and is usually deployed either once a year (annually), when the baseline prevalence is between 20 and 50%, or twice a year (biannually) when the prevalence is above 50% [7,8]. For over a decade, the World Health Organization (WHO) has coordinated the annual distribution of over 600 million albendazole/mebendazole medicines donated by pharmaceutical companies such as GlaxoSmithKline and Johnson & Johnson [7]. 

Nigeria remains the most endemic country for STH in sub-Saharan Africa, with records of all the four common STH species [1,9]. The country has a total of 774 implementation units (IU) where PC programs are implemented. These IUs were mapped in 2014 to establish prevalence estimates needed to recommend PC thresholds. Only 583 (75.3%) of the IUs were endemic and have subsequently benefited from PC since 2014 with significant support from the WHO, UNICEF and partner organizations such as Mission to save the helpless (MITOSATH), Sightsavers, and the AMEN foundation, among others [10,11]. Routine information on prevalence, intensity, and associated risk factors from these IUs have become invaluable to control programmers as they monitor PC progress, optimize resource allocation, and address emerging needs. However, there are limited investments in understanding the current prevalence and associated risk factors in IUs previously classified to be non-endemic. In line with the specific targets set out for eliminating STH in the 2020–2030 NTD road map [2,7,8], it is therefore imperative to provide updated prevalence and intensity estimates in non-endemic IUs to support timely decision-making and implementation, especially in scenarios where transmission dynamics are changing. 

Here, we present findings from an analysis of epidemiological data on soil-transmitted helminth infection obtained during a recent mapping exercise in Borgu, one of the non-endemic IUs for STHs in the northcentral region of Nigeria. The aims of this study were to estimate the current prevalence and intensity of STH and identify factors associated with transmission among school-aged children. 

## 2. Materials and Methods

### 2.1. Ethical Statement and Considerations

The study protocol was approved by the ethical review board of the Nigerian Institute of Medical Research (IRB/18/042). Requisite approvals and field permits were also obtained from the three study states prior to the commencement of data collection. Permissions were also obtained from school authorities and parents of the study participants. Only children who were above 6 years of age and whose parents consented to their participation through a written informed consent were invited to participate in the study. Children’s assent was obtained verbally and also documented through an assent form in the presence of the parent or legal guardian. Unique identifiers and a password-protected database were also used to ensure anonymity and confidentiality through the study procedures. The study procedures were implemented in accordance with the ethical standards of the Helsinki Declaration (1964, amended most recently in 2008) of the World Medical Association. 

### 2.2. Study Area

This study was carried out in five communities (Monai, Tamanai, Koro, Musarwa, and Yuna) around Kainji Dam in Borgu LGA of Niger state. Niger is located in the northcentral region of Nigeria and has a total of 25 administrative units, known as local government areas (LGAs). Borgu is one of the 25 LGAs and is located in the northwestern part of the state (Figure 1). The demographics of Borgu have been published previously [12], with a total land area of 16,219 km^2^ and an estimated population of 115,000 [12]. The predominant human occupations are farming, fishing, and trading. There are limited interventions in terms of piped-borne water and latrine facilities. This LGA is considered non-endemic for STH; however, it is endemic for lymphatic filariasis (LF), another debilitating neglected tropical disease that affects the lymphatic system, causing clinical manifestations such as hydrocoele and lymphoedema. The LGA has thus benefitted from 10 annual rounds of community-based LF MDA with Ivermectin and Albendazole.

### 2.3. Study Design

This study employed a cross-sectional sampling design involving questionnaire administration and collection of stool samples from school-aged children between ages 6 and 16 years. Our study design and sample size estimations followed the WHO guidelines of recruiting pupils during helminthiasis survey [13,14].

### 2.4. Selection of Communities

Preliminary advocacy and sensitization visits were made to the selected communities prior to epidemiological investigation. The study was conducted in the month of April 2019 and involved 4 distinct phases: (1) advocacy and sensitization; (2) questionnaire administration; (3) sample collection and laboratory examination; and (4) treatment of all recruited persons.

### 2.5. Sample Size Determination and Selection of Study Participants

Our study size estimations followed WHO guidelines of recruiting a minimum of 50 pupils (above age 5) per school during helminthiasis survey [13,14]. We adjusted the sample size by 60% in each of the communities to cater for non-enrolled school-aged children. As such, we targeted a minimum of 400 children across the five selected communities. However, the recruitment of participants was in variant with the estimated sample size for some communities as a result of number of eligible participants who presented themselves for the study. Summarily, a total of 410 children were enrolled: 133 from Monai, 79 from Tamanai, 124 from Koro, 36 from Musarwa, and 38 from Yuna. There was only one school in each community, and the enrollment of participants and collection of samples took place at a central point in the community, which was provided by the community leader. Each point had a secluded space for administering study questionnaires and sorting samples before transporting them to the laboratory. The number of consenting participants varied across the communities, hence giving an unequal number of persons recruited. Only children who had not received albendazole/mebendazole medicines from ongoing LF PC programs within the last 6 months were recruited into the study.

### 2.6. Questionnaire Administration

A simple questionnaire was used to collect demographic (age, gender, and parental occupation) and attitudinal (access to latrine and walking barefooted) data from the participants. The questionnaire was first designed in the English language and then translated to the Hausa language. Prior to administration of the questionnaire, recruited participants who had completed informed assent forms were assigned unique identification numbers. Each participant’s unique identification number was used to allocate a pre-labeled sterile stool specimen bottle. 

### 2.7. Collection of Stool Samples

Participants were provided with one sterile specimen bottle pre-labeled with their unique identification number, an applicator stick, a plain sheet of paper, and tissue paper to clean their anus. Participants were instructed to defecate on the plain sheet of paper and use the applicator stick to transfer a fresh portion into the specimen bottle. Specimen bottles were retrieved within 1 h of distribution, but collection was between 10 h and 14 h. All participants were treated with albendazole as an immediate benefit of the research investigation. 

### 2.8. Parasitological Assessment of Stool Samples

All collected stool samples were sorted and transported in iceboxes for processing within 2 h of collection to the Parasitology laboratory located in Borgu General Hospital. The specimens were processed using the Kato–Katz technique. Two thick smears were prepared from a single stool sample and allowed to clear for 30 min before microscopic examination for ova of STH parasites. The fields were also re-examined and counterchecked by another microscopist. A participant was considered infected if there was an egg count recorded on both sheets of the two microscopists who examined the smears. The endemicity was classified based on the aggregated prevalence in each community. Communities were classified as non-endemic when prevalence was below 20%, lowly endemic when prevalence was between 20 and 49.9%, and highly endemic when prevalence was above 50% [6]. Furthermore, the intensity of infection, i.e., egg count per 1 g of stool, was also determined by multiplying the number of eggs counted per smear by 24. Intensities for ova of *Ascaris lumbricoides* were categorized as low when the resulting EPG was between 1–4999, moderate when the EPG was between 5000 and 49,999, and high-intensity when the EPG was above 50,000 [6]. Similarly, for hookworms, intensity was classified as low when the EPG was between 1–1999, moderate when the EPG was between 2000 and 3999, and high when the EPG was above 4000 [6].

### 2.9. Data Management and Analysis 

Data obtained from this study Appendix A were imported and analyzed in R. software version 4.3.2. Descriptive statistics including frequencies and percentages were used to describe the variables. Following this, we performed univariate chi-square statistics between the outcome variable (prevalence of STH) other explanatory variables (age, gender, parent occupation, access to latrine, and walking barefoot). Significant associations were established when *p* < 0.05. Furthermore, we performed a univariate logistic regression model with each of the explanatory variables against the outcome variable. Only variables with *p* < 0.2 in the univariate model were included in the multivariate logistic model. Regression estimates in the form of odds ratios (OR) and 95% confidence intervals (CI) were calculated in the univariate model, and these estimates were adjusted in the multivariate model. For both models, the significance level was established as *p* < 0.05. 

## 3. Results

### 3.1. Demographic Characteristics of Study Participants

Table 1 shows the gender and age distribution of the study participants. A total of 410 school-aged children from five communities: Monai (133, 32.4%), Tamanai (79, 19.3%), Koro (124, 30.2%), Musawa (36, 8.8%), and Yuna (38, 9.3%) were enrolled in this study. The majority of the participants were males (237, 57.8%) when compared to females (173, 42.2%). By age category, the majority of the participants were between ages 6 and 8 years (170, 41.5%), followed by those between 9 and 11 years of age (119, 29.0%), 12–14 years (110, 26.8%), and 15–17 years (11, 2.7%). However, the majority of the participants recruited in Yuna (65.8%) and Tamanai (54.4%) were within the age categories of 12–14 and 9–11 years, respectively. The majority of the participants had parents who worked as farmers (149, 36.3%), civil servants (123, 30.0%), and fishermen (53, 12.9%). More than two-thirds (284, 69.3%) of the participants did not have access to latrines. Open defecation was more common across the study communities except in Tamanai (59, 74.7%). Furthermore, about 47% of the participants walked barefoot. However, this practice was less common in the communities of Monai (46, 34.6%) and Tamanai (11, 13.9%). The distribution of participants based on gender, age, parent’s occupation, access to latrines, and the practice of walking barefoot was significantly different across the study communities (*p* = 0.00). 

### 3.2. Prevalence and Intensity of Soil-Transmitted Helminths among the Study Participants

Table 2 shows the prevalence of soil-transmitted helminths among the study participants. Of the 410 participants examined, a total of 36 (8.8%) were infected with both *Ascaris lumbricoides* (16, 3.9%) and hookworm (20, 11.3%). The prevalence of STH varied across the communities, with the highest recorded in Koro (14.5%), followed by Tamanai (12.7%), Musawa (5.6%), and Monai (4.5%) (Table 2). Only three of the five communities had participants infected with *Ascaris lumbricoides*: Koro (8.9%), Tamanai (3.8%), and Monai (1.5%). The communities reported a low intensity of infection, with EPG counts ranging from 24 to 72 and having a mean of 40. Furthermore, only four of the five communities had participants infected with hookworm: Tamanai (8.9%), Koro (5.6%), Musawa (5.6%), and Monai (3.0%). The communities also reported a low intensity of infection, with EPG counts ranging from 24 to 1344 and having a mean of 419.

### 3.3. Association between Soil-Transmitted Helminths and Other Sociodemographic Variables

Table 3 shows the association between STH infections and other demographic variables. STH parasites were more common in Koro (14.5%) and Tamanai’s (12.7%) communities. The prevalence of STH was significantly associated with communities (*p* = 0.008), and the univariate logistic regression model shows that Tamanai (OR: 0.33 (95% CI: 0.11–0.92)) and Koro (OR: 0.28 (95% CI: 0.1–0.69)) have a significantly reduced odds of being endemic with STH (*p* < 0.05) when compared to Monai’s community. However, for gender, males had a prevalence of 10.4% compared to females with 7.6%, but there was no significant association (*p* = 0.41). The univariate regression model also showed that females were 1.4 times more likely to become infected compared to males; however, this finding was also not significant (OR: 1.41 (95% CI: 0.71–2.82)). STH prevalence was more common among participants between ages 9–11 years (11.8%) and 12–14 years (10.9%). However, there was no significant association between age and infection (*p* = 0.21). The univariate model further reveals those participants within the age group of 9–11 years had significantly lower odds of becoming infected (OR: 0.42 (95% CI: 0.17–0.99)) compared to children between ages 6–8 (*p* = 0.05). Furthermore, children whose parents were traders (20.7%), artisans (16.7%), miners (13.8%), civil servants (10.6%), farmers (6.0%), or fishermen (1.9%) were infected with STH. Occupation was significantly associated with STH infection (*p* = 0.038), and children whose parents were traders had significantly reduced odds of infection (OR: 0.25 (95% CI: 0.08–0.79)) compared to those whose parents were farmers (*p* = 0.01). By practice, having access to latrines and walking barefoot were not associated with STH infection (95% CI included one and *p* > 0.05). Our multivariate regression models did not find any significant covariates for STH infection (Table 3), and individual models for *Ascaris* and hookworms are shown in Table A1 (95% CI included one and *p* > 0.05).

## 4. Discussion

The 2030 elimination targets for STH specifies that the prevalence of moderate- and heavy-intensity infections should be below 2% [2]. This target requires concerted efforts of control programs, especially in terms of optimizing PC programs in already-identified transmission zones, and also monitoring transmission patterns in areas previously adjudged to be non-endemic. The reason for the latter can be linked with emerging reports where highly endemic hotspots were found in IUs previously thought to be moderately endemic [15]. Routine provision of information on prevalence, intensity, and associated risk factors from non-endemic and moderately endemic IUs have thus become invaluable to address emerging threats, optimize control/elimination strategies, and efficiently allocate resources. This study therefore analyzed epidemiological data from a recent mapping exercise for soil-transmitted helminthiasis in Borgu, one of the non-endemic IUs for STH in the northcentral region of Nigeria, with the aim of monitoring changes in prevalence and intensity estimates, further probing into factors associated with transmission. 

Borgu was mapped in 2013 by the Federal Ministry of Health with an overall baseline prevalence of 18.3% recorded across five randomly selected communities [16,17]. The IU was classified to be non-endemic in 2014, and has benefitted from one PC with albendazole in the last seven years [16]. Our findings after a 5-year non-intervention period show a significantly reduced prevalence of 8.8%, which is about a 51.9% change. Furthermore, all the infected participants (36 out of 410) had a low intensity of infection, which is in line with the elimination targets of the country. Although the disaggregated baseline intensity estimate report for the IU is not publicly available to allow comparison, only three of the twenty-five IUs in Niger state had heavy intensity in 2013, and nineteen of them had low intensity [17]. It is therefore important to maintain this threshold as the STH elimination program in the state works towards a significant reduction of intensities across the highly endemic IUs. This endeavor should be complemented with periodic epidemiological assessments involving parasite detection and the estimation of associated risk factors.

Despite the low prevalence and intensity estimates, our univariate regression model found significantly reduced odds in some communities (about 21–25% reduced odds) and among children whose parents were traders (20 times lower) when compared with those whose parents were farmers. The risk of transmission of STH parasites has been reported to be higher among farmers due to their increased exposure to soils that are likely to have been contaminated with the infective stages of the parasites [18,19,20]. This risk is particularly heightened in areas like those in these studies where about 69% of the children do not have access to latrine facilities and 45% of them walk barefoot. Investments in the provision of water, sanitation, and hygiene (WASH) resources and health education are therefore vital in the process of sustaining the low endemic levels reported [21,22]. 

Furthermore, there are concerns that occupational risks may predispose adults to STH infections, making them serve as reservoirs for STH infections to children [23,24,25,26,27]. However, these adult populations are currently not included in PC programs targeted at STH [27]. There is thus an appreciable growing body of evidence that the consideration of these adult populations in STH PC programs would complement efforts targeted at reducing exposure among the children [23,24,25,26,27]. However, in areas where STH and lymphatic filariasis (LF) are co-endemic, there are records of complimentary benefits as albendazole and ivermectin tablets are distributed during PC for LF [28]. Residents above 5 years of age who benefit from this LF PC program are therefore in-directly treated for STH. Borgu is endemic for lymphatic filariasis and has benefitted from over 10 rounds of PC targeted at LF [16]. This is likely the reason for the 51.9% change in STH prevalence reported in the study area, despite the lack of specific PC for STH. The complementary benefits of PC targeted at LF on STH prevalence have been extensively discussed and reported [28,29,30]. However, it is important to consider and address the inherent technicalities of implementing such joint co-implementation, especially on issues around differences in strategies of mapping endemic hotspots and administering the medicines, i.e., community-based PC intervention for LF versus school-based intervention for STH [29]. This also applies to the promotion of other complementary interventions, including the ongoing advocacy for WASH [21,22] and other health educational tools that have been proven to have potential for reducing STH prevalence [31,32]. 

## 5. Conclusions

In this study, we report a significantly reduced prevalence and intensity for soil-transmitted helminths in an implementation unit previously mapped in 2013 and adjudged to be non-endemic. Despite the low endemicity, our findings revealed poor access to latrines and footwear usage. Farming was also associated with increased risk, and the ongoing preventive chemotherapy program for lymphatic filariasis could be responsible for the huge reduction in prevalence and intensity estimates. It is therefore important to invest in monitoring transmission dynamics in other non-endemic areas to arrest emerging threats by promoting complementary PC interventions and investing in the provision of WASH facilities and other health educational tools that have been proven to have potential for reducing STH prevalence.

## Figures and Tables

**Figure 1 pathogens-12-00491-f001:**
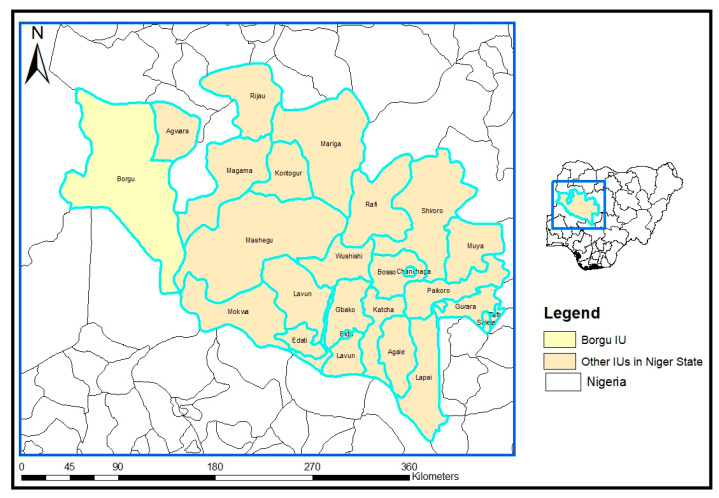
Map of Niger state showing the 25 implementation units, with Borgu and Nigeria as a highlight and inset, respectively.

**Table 1 pathogens-12-00491-t001:** Demographic characteristics of the study population.

	Communities						
	Monai (*n* = 133)	Tamanai (*n* = 79)	Koro (*n* = 124)	Musawa (*n* = 36)	Yuna (*n* = 38)	Total (*n* = 410)	*p*-Value
**Sex**							
Female	56 (42.1)	41 (51.9)	56 (45.2)	16 (44.4)	7 (10.5)	173 (42.2)	0.000
Male	77 (57.9)	38 (48.1)	68 (54.8)	20 (55.6)	34 (89.5)	237 (57.8)	
**Age group in years**							
6–8	95 (71.4)	5 (6.3)	41 (33.1)	22 (61.1)	7 (18.4)	170 (41.5)	0.001
9–11	18 (13.5)	43 (54.4)	38 (30.6)	14 (38.9)	6 (15.8)	119 (29.0)	
12–14	20 (15.0)	26 (32.9)	39 (31.5)	0 (0)	25 (65.8)	110 (26.8)	
15–17	0 (0)	5 (6.3)	6 (4.8)	0 (0)	0 (0)	11 (2.7)	
**Parent’s occupation**							
Farming	49 (36.8)	20 (25.3)	39 (31.5)	20 (55.6)	21 (55.3)	149 (36.3)	0.001
Fishing	18 (13.5)	0 (0)	2 (1.6)	16 (44.4)	17 (44.7)	53 (12.9)	
Civil servant	62 (46.6)	29 (36.7)	32 (25.8)	0 (0)	0 (0)	123 (30.0)	
Trading	1 (0.8)	15 (19.0)	13 (10.5)	0 (0)	0 (0)	29 (7.1)	
Artisan	0 (0)	12 (15.2)	6 (4.8)	0 (0)	0 (0)	18 (4.4)	
Miner	0 (0)	0 (0)	29 (23.4)	0 (00	0 (00	29 (7.1)	
Driver	3 (2.3)	3 (3.8)	3 (2.4)	0 (0)	0 (0)	9 (2.2)	
**Access to latrine**							
Yes	29 (21.8)	59 (74.7)	38 (30.6)	0 (0)	0 (0)	126 (30.7)	0.001
No	104 (78.2)	20 (25.3)	86 (69.4)	36 (100)	38 (100)	284 (69.3)	
**Walking barefoot**							
Yes	46 (34.6)	11 (13.9)	73 (58.9)	34 (94.4)	28 (73.7)	192 (46.8)	0.001
No	87 (65.4)	68 (86.1)	51 (41.1)	2 (5.6)	10 (26.3)	218 (53.2)	

**Table 2 pathogens-12-00491-t002:** Prevalence and intensity of soil-transmitted helminths.

	Overall STH		*A. lumbricoides*		Hookworm	
Communities	NE	NI (%)	NI (%)	Mean EPG ± SD (Min, Max)	NI (%)	Mean EPG ± SD (Min-Max)
Monai	133	6 (4.5)	2 (1.5)	48 ± 33.94 (24, 72)	4 (3.0)	630 ± 421.37 (24, 960)
Tamanai	79	10 (12.7)	3 (3.8)	40 ± 13.86 (24, 48)	7 (8.9)	572.57 ± 383.36 (240, 1344)
Koro	124	18 (14.5)	11 (8.9)	39.27 ± 16.18 (24, 72)	7 (5.6)	250.29 ± 287.28 (24, 840)
Musawa	36	2 (5.6)	-	-	2 (5.6)	48 ± 0.0 (0, 48)
Yuna	38	0 (0)	-	-	0 (0)	-
**Total**	410	36 (8.8)	16 (3.9)	40 ± 17.0 (24, 72)	20 (4.9)	419 ± 379 (24, 1344)

NE: Number of children examined; NI: number of children infected; EPG: eggs per gram of feces; SD: standard deviation; Min: minimum number of egg count; Max: maximum number of egg count. Categories of endemicity based on overall prevalence; a: non-endemic when prevalence is below 20%; b: low endemicity when prevalence is between 20 and 49.9%; high endemicity when prevalence is above 50% [6]. Categories of intensity based on EPG for *Ascaris lumbricoides*; low intensity when EPG is between 1–4999; moderate intensity when EPG is between 5000 and 49,999; high intensity when EPG is above 50000 [6]. Categories of intensity based on EPG for hookworms; low intensity when EPG is between 1–1999; moderate intensity when EPG is between 2000 and 3999; high intensity when EPG is above 4000 [6].

**Table 3 pathogens-12-00491-t003:** Association between soil-transmitted helminths and other sociodemographic variables.

Covariates	Frequency (%)	Positives (%)	Negatives (%)	*p*-Value	OR (95% CI)	*p*-Value	AOR (95% CI)	*p*-Value
	*n* = 410	*n* = 36 (8.8)	*n* = 374 (91.2)					
**Communities**								
Monai	133 (32.4)	6 (4.5)	127 (95.5)	0.008 *	REF	-	REF	-
Tamanai	79 (19.3)	10 (12.7)	69 (87.3)		0.33 (0.11–0.92)	0.04 *	0.59 (0.16–2.03)	0.41
Koro	124 (30.2)	18 (14.5)	106 (85.5)		0.28 (0.1–0.69)	0.01 *	0.39 (9.12–1.14)	0.09
Musawa	36 (8.8)	2 (5.6)	34 (94.4)		0.8 (0.18–5.65)	0.79	0.53 (0.09–4.26)	0.5
Yuna	38 (9.3)	0 (0)	38 (100)		-	-	-	-
**Gender**								
Male	173 (42.2)	18 (7.6)	155 (89.6)	0.41	REF	-	-	-
Female	237 (57.8)	18 (10.4)	219 (92.4)		1.41 (0.71–2.82)	0.32	-	-
**Age group in years**								
Age (6–8)	170 (41.5)	9 (5.3)	161 (94.7)	0.21	REF	-	REF	-
Age (9–11)	119 (29.0)	14 (11.8)	105 (88.2)		0.42 (0.17–0.99)	0.05 *	0.56 (0.21–1.5)	0.25
Age (12–14)	110 (26.8)	12 (10.9)	98 (89.1)		0.46 (0.18–1.12)	0.09	0.54 (0.19–1.45)	0.22
Age (15–17)	11 (2.7)	1 (9.1)	10 (90.9)		0.56 (0.09–10.8)	0.60	0.93 (0.13–19.1)	0.95
**Parental occupation**								
Farming	149 (36.3)	9 (6.0)	140 (94.0)	0.038 *	REF	-	REF	-
Fishing	53 (12.9)	1 (1.9)	52 (98.1)		3.34 (0.61–62.4)	0.26	2.13 (0.31–43.2)	0.51
Civil servant	123 (30.0)	13 (10.6)	110 (89.4)		0.54 (0.22–1.31)	0.18	0.61 (0.23–1.56)	0.31
Trading	29 (7.1)	6 (20.7)	23 (79.3)		0.25 (0.08–0.79)	0.01 *	0.41 (0.12–1.41)	0.14
Artisan	18 (4.4)	3 (16.7)	15 (83.3)		0.32 (0.08–1.56)	0.11	0.54 (0.13–2.82)	0.42
Miner	29 (7.1)	4 (13.8)	25 (86.2)		0.4 (0.12–1.57)	0.15	0.74 (0.19–3.12)	0.66
Driver	9 (2.2)	0 (0)	9 (100)		-	-	-	-
**Access to latrine**								
Yes	126 (30.7)	9 (7.1)	117 (92.9)	0.55	REF	-	-	-
No	284 (69.3)	27 (9.5)	257 (90.5)		0.73 (0.32–1.55)	0.44	-	-
**Walking barefoot**								
Yes	192 (46.8)	17 (8.9)	175 (91.1)	0.99	REF	-	-	-
No	218 (53.2)	19 (8.7)	199 (91.3)		1.02 (0.51–2.02)	0.96	-	-

N: number of persons examined; OR: odds ratio; CI: confidence interval; REF: reference category; * significant difference at 95%.

## Data Availability

The datasets used and/or analyzed during the current study are available as Appendix A.

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
