# Peer review of "Monitoring the Status of Soil-Transmitted Helminthiases in Non-Endemic Implementation Units: A Case Study of Borgu in Northcentral Nigeria"

_pathogens, 2023, doi:10.3390/pathogens12030491_

Round 1

Reviewer 1 Report

The study of the authors is devoted to topical issues. This research deals with distribution of soil transmitted helminthiases among human population in Borgu (Nigeria). The study of the authors is important due to the importance of control, treatment and preventing dangerous human diseases.

Manuscript is suitable to be published in Pathogens but I have minor remarks, which will undoubtedly improve the article.

You have identified eggs of several species of nematodes in your study, then “soil-transmitted helminthiases” or “soil-transmitted helminth infections” should be used. Please check throughout the text.

There is only one image in the article and it is only a map of the study site. The authors used the P environment for the calculations. Several qualitative illustrations could have been made to support the results obtained, rather than using only tables for this purpose. This would have been much better.

and a few more comments:

1. Please use the full names of parasitological parameters throughout the text: “prevalence of infection, intensity of infection.

2. According International Code of Zoological Nomenclature (ICZN), at the first mention of species, its full Latin name with the author and year of description should be given; in relation all species of parasites and their hosts (for example, Ascaris lumbricoides Linnaeus, 1758, Trichuris trichiura (Linnaeus, 1771), etc.). And it is desirable to add higher order taxa such as family and order.

3. Lines 40,41 – All species are nematodes. Please, use this term.

4. Line 99 – “human occupation”

5. In figure 1, the location of the small square (rectangle) on the map of Nigeria is somewhat different from the large square. Please correct.

6. “first bottle”? May be “sterile bottle” or “specimen bottle”?

7. Names of the subsection 3.2. and Table 2 should be corrected. Not helminthiases but helminths are transmitted through the soil. So it's better like this: “Prevalence and intensity of infection of soil transmitted nematodes among the study participants”

8. In the table it is better to use the Latin names of helminths. Not “Hookworm” but better Ancylostoma sp.” (if you haven't identified to the species level).

9. Lines 219 and 243 – May be better “Association between infection with soil-transmitted nematodes and socio-demographic variables”?

10. Lines 241, 317 – italics for Latin name.

11. In Discussion there are no references to own results (tables) (for example, Lines 264, 265). Or is this a comparison with results of other authors? Please, clarify.

12. Line 317 – “… for Ascaris lumbricoides and Ancylostoma sp.”

Author Response

Reviewer 1:

General Comment: The study of the authors is devoted to topical issues. This research deals with distribution of soil transmitted helminthiases among human population in Borgu (Nigeria). The study of the authors is important due to the importance of control, treatment and preventing dangerous human diseases.Manuscript is suitable to be published in Pathogens but I have minor remarks, which will undoubtedly improve the article.

Comment 1: You have identified eggs of several species of nematodes in your study, then “soil-transmitted helminthiases” or “soil-transmitted helminth infections” should be used. Please check throughout the text.

Response 1:  Thank you very much for your comment. We have revised the entire text and replaced with “soil-transmitted helminthiases”

Comment 2; There is only one image in the article and it is only a map of the study site. The authors used the P environment for the calculations. Several qualitative illustrations could have been made to support the results obtained, rather than using only tables for this purpose. This would have been much better.

Response 2: We agree with the reviewer on this comment. We plotted charts for some demographic variables at the initial stage of our results presentation, but we realized they were less informative and decided to use tables.

  1. Please use the full names of parasitological parameters throughout the text: “prevalence of infection, intensity of infection.

Response 3: Thank you very much. The manuscript text has been revised as suggested

  1. According International Code of Zoological Nomenclature (ICZN), at the first mention of species, its full Latin name with the author and year of description should be given; in relation all species of parasites and their hosts (for example, Ascaris lumbricoidesLinnaeus, 1758, Trichuris trichiura (Linnaeus, 1771), etc.). And it is desirable to add higher order taxa such as family and order.

Response 4: Thank you very much for this comment. We have revised the text accordingly, please see line 39-44 below:

Most human STH infections are from four parasitic helminths species whose development are dependent on the soil environment; Ascaris lumbricoides (Linnaeus, 1978) known as  roundworm, Trichuris trichiura (Linnaeus,  1771) known as whipworm, Necator americanus (Stiles, 1902) and Ancylostoma duodenale (Dubini, 1843) both known as hookworms. [3].

  1. Lines 40,41 – All species are nematodes. Please, use this term.

Response 5: Thank you very much for this comment. We have revised the text accordingly,

  1. Line 99 – “human occupation”

Response 6: Thank you very much for this comment. We have revised the text accordingly,

  1. In figure 1, the location of the small square (rectangle) on the map of Nigeria is somewhat different from the large square. Please correct.

Response 7: Thank you very much for this comment. We have replaced the figure with a new one. We reduced the size of the blue square and also included the subunits in the zoomed map.

  1. “first bottle”? May be “sterile bottle” or “specimen bottle”?

Response 8: Thank you very much for this comment. We have revised the text accordingly,

  1. Names of the subsection 3.2. and Table 2 should be corrected. Not helminthiases but helminths are transmitted through the soil. So it's better like this: “Prevalence and intensity of infection of soil transmitted nematodes among the study participants”

Response 9: Thank you very much for this comment. We have revised the text accordingly. However, we maintained the used of helminths instead of nematodes, to allow conformity with earlier usage and the acronym “STH”

  1. In the table it is better to use the Latin names of helminths. Not “Hookworm” but better “Ancylostoma sp.” (if you haven't identified to the species level).

Response 10:  We thank you very much for this comment, but we will like to use Hookworm rather than Ancylostoma sp., to avoid confusion since we didn’t identify the worms to the species level. This has been a customary practice by several authors, and is especially useful for first-time readers. Your consideration will be appreciated.

  1. Lines 219 and 243 – May be better “Association between infection with soil-transmitted nematodes and socio-demographic variables”?

Response 11: Thank you very much for this comment. We have revised the text accordingly. However, we maintained the used of helminths instead of nematodes, to allow conformity with earlier usage and the acronym “STH”

  1. Lines 241, 317 – italics for Latin name.

Response 12: Thank you very much for this comment. We have revised the text accordingly.

  1. In Discussion there are no references to own results (tables) (for example, Lines 264, 265). Or is this a comparison with results of other authors? Please, clarify.

Response 13: Thank you very much for this comment. This line of discussion was actually in comparison with baseline findings, which has been referenced as [16].

  1. Line 317 – “… for Ascaris lumbricoides and Ancylostoma sp.”

Response 14: We thank you very much for this comment, but we will like to use Hookworm rather than Ancylostoma sp., to avoid confusion since we didn’t identify the worms to the species level. This has been a customary practice by several authors, and is especially useful for first-time readers. Your consideration will be appreciated.

Reviewer 2 Report

Major Comments :

The paper covers a study to monitor the status of STH in Borgu, a non-implementation unit in the North-central part of Nigeria. Overall, the paper is very well written and the recommended methodology of WHO has also been followed. The language and grammar are fine but some minor editing is required to address the issues related to tense and punctuation etc. In addition to these minor corrections in the text, calculations and tables are also required for the finalization of the paper.

The paper can be considered for publication after the suggested minor changes.

Minor/Specific Comments  :

1.     Abstract :

Lines 20-21: The authors state,’ An overall prevalence of 8.8% was recorded for STH infection, which corresponds to 108% change from the 18.3% reported in 2013’. Please check the calculation. The actual decline is 51.9%.

2.     Introduction :

·        Lines 42,51,75: Pl check grammar.

· Ensure proper punctuation e.g., commas before and after ‘Therefore’, and comma after ‘However’ as these have been used several times in the paper.

3.     Material and Methods :  

Sample size determination and selection of study participants :

Please include the total number of schools available and the number included in each of the five study areas/localities i.e., Monai, Tamanai, Koro, Musarwa and Yuna.

Lines 130-131: Since PC is not given to the children in non-endemic areas why and how was Albendazole/mebendazole given to some school children excluded from the study? Please clarify in the paper otherwise it seems to be contradictory to the statement given in the introduction.

4.     Results :

Line 189: Check the figure for fisherman here as well as in table 1(Koro)

Table 1: Please check figure 291.6 under the column Koro for Fishing occupation as well as the column total. Something is wrong.

Line 199: Pl check this. Ascaris lumbricoides affects humans and causes the disease ascariasis. In case an animal species was found, please clarify.   If you mean to say  ' were infected with both Ascaris lumbricoides and hookworm',  then pl. delete species from this sentence to convey the intended meaning.

224: Replace been with being.

Lines 234 and 236: Pl check these lines. Both appear contradictory for children of traders.

Discussion :

Lines 262-265,296: The change amounts to 51.9126% decrease only and not 108%. Please check this in the entire paper.

References :

Please ensure that the journal format is followed for all references.

Reference 4 : Pl check the relevance of this reference. This reference is out of context.

Author Response

Reviewer 2

Major Comments: The paper covers a study to monitor the status of STH in Borgu, a non-implementation unit in the North-central part of Nigeria. Overall, the paper is very well written and the recommended methodology of WHO has also been followed. The language and grammar are fine but some minor editing is required to address the issues related to tense and punctuation etc. In addition to these minor corrections in the text, calculations and tables are also required for the finalization of the paper.

General Comment: The paper can be considered for publication after the suggested minor changes.

Minor/Specific Comments :

  1. Abstract:

Lines 20-21: The authors state,’ An overall prevalence of 8.8% was recorded for STH infection, which corresponds to 108% change from the 18.3% reported in 2013’. Please check the calculation. The actual decline is 51.9%.

Response 1:(We are very grateful for the reviewer for this comment. We have revised the text appropriately

  1. Introduction:
  • Lines 42,51,75: Pl check grammar. Ensure proper punctuation e.g., commas before and after ‘Therefore’, and comma after ‘However’ as these have been used several times in the paper.

Response 2:( Thank you very much for this comment. We have revised the text accordingly.

  1. Material and Methods :  

Sample size determination and selection of study participants :

Please include the total number of schools available and the number included in each of the five study areas/localities i.e., Monai, Tamanai, Koro, Musarwa and Yuna.

Response 3:( Thank you very much for this comment. We have revised the text accordingly.

Summarily, a total of 410 children were enrolled; 133 from Monai, 79 from Tamanai, 124 from Koro, 36 from Musarwa and 38 from Yuna. There was only one school in each community, and the enrollment of participants and collection of samples took place at a central point in the community, which was provided by the community leader

  1. Lines 130-131: Since PC is not given to the children in non-endemic areas why and how was Albendazole/mebendazole given to some school children excluded from the study? Please clarify in the paper otherwise it seems to be contradictory to the statement given in the introduction.

Response 4:( Thank you very much for this comment. We have revised the text accordingly. We have clarified this in the introductory section of the manuscript. Although, the LGA is non-endemic for STH, and hence not benefitting from STH-specific MDA programs, however, the LGA is endemic for LF, and community based LF MDA programme using Ivermectin and Albendazole is ongoing.

Please see line 102-106

This LGA is considered non-endemic for STH, however, it is endemic for lymphatic filariasis (LF), another debilitating neglected tropical diseases that affects the lymphatic system causing clinical manifestations such as hydrocoele and lymphoedema. The LGA has thus benefitted from 10 annual rounds of community-based LF MDA with the Ivermectin and Albendazole

                Please see Line 139-141

Only children who had not received albendazole/mebendazole medicines from ongoing LF MDA programme within the last 6 months were recruited into the study.

  1. Results :

Line 189: Check the figure for fisherman here as well as in table 1(Koro). Table 1: Please check figure 291.6 under the column Koro for Fishing occupation as well as the column total. Something is wrong.

Response 5:( Thank you very much for this comment. We have revised the text in the table 2(1.6), instead of 291.6). the 9 was meant to be an open parenthesis. The total is correct and the text description is correct. We thank the reviewer for this observation.

  1. Line 199: Pl check this. Ascaris lumbricoidesaffects humans and causes the disease ascariasis. In case an animal species was found, please clarify.   If you mean to say  ' were infected with both Ascaris lumbricoides and hookworm',  then pl. delete species from this sentence to convey the intended meaning.

Response 6:( Thank you very much for this comment. We have revised the text by deleting species

  1. 224: Replace been with being.

Response 7:( Thank you very much for this comment. We have revised the text

  1. Lines 234 and 236: Pl check these lines. Both appear contradictory for children of traders.

Response 8:( Thank you very much for this comment. We have revised the text accordingly. Please see below

Furthermore, children whose parents were traders (20.7%), artisans (16.7%), miners (13.8%), civil servants (10.6%), farmers (6.0%) and fishermen (1.9%) were infected with STH.

Discussion :

Lines 262-265,296: The change amounts to 51.9126% decrease only and not 108%. Please check this in the entire paper.

Response: We are very grateful for the reviewer for this comment. We have revised the text appropriately

References :

  1. Please ensure that the journal format is followed for all references.

                        Response: We have formatted the references to the Journal's style

  1. Reference 4 : Pl check the relevance of this reference. This reference is out of context

Response: We used this reference to re-inforce the morbidities associated with STH, especially those related with iron deficiency, and cognitive malnutrition.